# The Impact of Mental Fatigue on the Accuracy of Penalty Kicks in College Soccer Players

**DOI:** 10.3390/sports13080259

**Published:** 2025-08-07

**Authors:** Qingguang Liu, Ruitian Huang, Zhibo Liu, Caiyu Sun, Linyu Qi, Antonio Cicchella

**Affiliations:** 1International College of Football, Tongji University, Shanghai 200092, China; qingguang@tongji.edu.cn (Q.L.); 2155062@tongji.edu.cn (R.H.); 2155001@tongji.edu.cn (Z.L.); 2434090@tongji.edu.cn (C.S.); 2434080@tongji.edu.cn (L.Q.); 2Department for Quality-of-Life Studies, University of Bologna, 47921 Rimini, Italy

**Keywords:** mental fatigue, shooting accuracy, soccer players, shooting time, soccer shooting performance

## Abstract

Purpose: To investigate the impact of mental fatigue on the shooting accuracy and movement timing in the instep kick of Asian high-level soccer players. Methods: Eight male collegiate soccer players (age 22.00 ± 0.93 years) were tested before and after mental fatigue induction. Mental fatigue was induced via a 30 min Stroop task. The effectiveness of fatigue induction was assessed using heart rate variability (HRV), a visual analog scale (VAS), rating of perceived exertion (RPE), and the Athlete Burnout Questionnaire (ABQ). Shooting performance was evaluated before and after mental fatigue using the Loughborough Soccer Shooting Test (LSST) and by evaluating timing by means of high-speed imaging. Results: Following mental fatigue induction, HRV decreased. Subjects’ motivation (VAS) to exercise significantly decreased (*p* < 0.001), while VAS mental fatigue level (*p* < 0.001) and mental effort level (*p* < 0.002) significantly increased. Significant differences were observed after completing the Stroop task for ABQ Emotional/Physical Exhaustion (*p* < 0.007), Reduced Sense of Accomplishment (*p* < 0.007), Sport Devaluation (*p* < 0.006), and overall burnout level (*p* < 0.002). LSST showed that the subjects’ left foot test scores (−4.13, *p* < 0.013), right foot test scores (−3, *p* < 0.001), and total scores (−3.16, *p* < 0.001) significantly decreased. Although movement times increased slightly after fatigue, they did not reach statistical significance. Conclusions: Mental fatigue significantly impairs the shooting accuracy of collegiate soccer players, as evidenced by decreased LSST scores. However, it has no significant effect on event duration during shooting execution. Mental fatigue affected decision-making but not shooting movement timing. More cognitively challenging tasks are more affected by mental fatigue.

## 1. Introduction

Mental fatigue is defined as a psychobiological state characterized by feelings of weariness and lack of energy, resulting from prolonged periods of high cognitive activity [1,2]. Research on soccer has confirmed that mental fatigue negatively impacts multiple athletic abilities [3].

Shooting is a critical determinant of match outcomes in soccer. Players engage in various technical and tactical activities to create scoring opportunities [4,5]. As matches progress, mental fatigue accumulates, impairing on-field performance [6,7]. Consequently, investigating the effect of mental fatigue and shooting technique in collegiate soccer players is highly relevant.

Regarding aerobic capacity, research demonstrates a negative impact of mental fatigue [3]. Futsal match analysis revealed that mentally fatigued players covered 3.3% less total distance than controls, with significant reductions in low-speed (7–9.9 km/h; −4.2%) and medium-speed (10–15.9 km/h; −10.7%) running distances. Conversely, mental fatigue appears to have minimal impact on anaerobic work capacity, maximal voluntary contraction strength, muscle explosive power, and sprinting performance [8].

Psychological cognitive protocols, such as the Stroop task, are the most widely used and validated methods for inducing mental fatigue in competitive sports research [9,10,11,12,13].

In soccer research, changes in technical execution under mental fatigue are primarily assessed using the Loughborough Soccer Passing Test (LSPT) and the Loughborough Soccer Shooting Test (LSST) [14]. A study with 14 male players employed a 30 min Stroop task versus neutral reading, followed by LSPT and LSST. Post-Stroop task, players exhibited significantly increased penalty time in the LSPT and a trend towards longer overall completion time. In the LSST, both shooting accuracy and speed significantly decreased [15]. Specifically, this study showed a reduction of more than 5 km/h in the shooting speed, and the average shot sequence time tended to be slower in the mental fatigue condition; however, differences were not significant. These findings indicate that mental fatigue impairs passing and shooting technique, a conclusion supported by other studies [16,17].

Decision-making involves the brain’s ability to perceive, process external information, and respond appropriately. Mental fatigue reduces brain working capacity [18], suggesting a corresponding decline in athletic decision-making. This was confirmed in a study with 12 Belgian professional players using a soccer-specific decision-making task [19]. After mental fatigue induction via a 30 min Stroop task, players watched match clips and made passing decisions. Evaluations based on passing direction, concealment, and threat level showed decreased reaction time and decision accuracy, indicating impaired passing decision-making. Recent studies consistently demonstrate that mental fatigue negatively impacts soccer players’ in-match decision-making [20,21].

Although research confirms mental fatigue’s detrimental effects on various aspects of soccer performance, to our knowledge, no studies have investigated its impact on the timing kinematics of kicking movements in an Asian population. Furthermore, the effect of the Stroop task on perceived exertion, burnout feelings, and heart rate variability (HRV) specifically in Asian soccer players remains unexplored. Therefore, the main aim of our study is to assess the influence of mental fatigue on the technique and timing of the instep kick, and, alongside, explore the effect of the Stroop test on the psychological dimensions of fatigue (perceived fatigue, burnout), and on HRV, in a cohort of high-level Asian soccer players.

## 2. Methods

### 2.1. Subjects

This study recruited eight male first-class soccer players from Tongji University (age: 22.00 ± 0.93 years; body weight: 69.50 ± 6.16 kg; height: 177.00 ± 5.78 cm). All participants were right-footed. In China, a “National First-Class Soccer Player” is an athlete who has attained a high level of proficiency and meets specific standards established by national sports authorities. As midfielders and forwards perform the majority of shots in actual matches, all recruited participants played in these positions.

Inclusion criteria were as follows: No injuries within the six months preceding the experiment, no use of medication, and refraining from training for at least two days prior to participation. Participants were instructed to ensure adequate rest the night before the experiment and to avoid high-load cognitive tasks. Prior to the study, all procedures were thoroughly explained, and written informed consent was obtained. As all participants resided in university dormitories, they shared similar lifestyle habits, including dining at campus canteens. The study was conducted in accordance with the Declaration of Helsinki and approved by the Ethics Committee of Tongji University (Approval Code: tjdxsr029).

### 2.2. Experimental Design

The LSST is a validated and reliable assessment tool for evaluating soccer players’ shooting skills [14,22,23]. Players sprinted between two cones positioned 5.5 m directly in front of the goal to take shots. Each subject performed ten shots (five with each foot), with a 30 s rest period between attempts. The performance score was calculated as the average of the total points accumulated from all shots that hit the target. Participants’ pre-test and post-test scores were recorded for subsequent comparison.

Shooting performance (instep kicks towards goal) was recorded using a 2D SH3-101 high-speed camera. (V1.0, Shenzhen Vision Intelligence Co., Ltd., Shenzen, China). Video analysis and timing calculations were performed using Shenshi Intelligent Motion Analysis software (V1.0, Shenzhen Vision Intelligence Co., Ltd., Shenzen, China). For both pre-test and post-test assessments, participants completed a shot filmed simultaneously from rear and side views. The camera operated at 250 frames per second with a resolution of 1280 × 1024 pixels. Timing is a basic measurement of the performance in the instep kick in soccer and has been proposed as a simple, time-efficient, and easily understandable tool for biomechanical performance assessment using basic measures [24]. This method has the advantage of not requiring complex calculations and being quickly usable on the soccer field. Being the instep kick, a movement essentially happening in the posterior–anterior direction, the errors introduced by the rotation can be significant; thus, we limited our analysis to the timing.

The following temporal variables were calculated before and after the mental fatigue intervention:FL/KF: Time from landing of the support foot to maximum knee flexion.FL/BC: Time from landing of the support foot to ball contact.FPRON: Time from landing of support foot to maximum foot inversion.TCT (Total Contact Time): Total time from landing of support foot to ball contact.Pre- to post-mental fatigue time differences were computed.

Among mental fatigue induction protocols in competitive sports research, the Stroop task is the most widely employed psychological cognitive task and serves as a representative paradigm [9,10,11,12]. This study utilized a modified Stroop protocol, integrating elements from Gantois et al. [11]. Following the pre-test of the Loughborough Soccer Shooting Test (LSST), participants performed a continuous 30 min Stroop task administered via E-Prime software (ver. 3.0.3.219) Immediately after completing the Stroop task, the following measures were collected to assess the successful induction of mental fatigue:Heart rate variability (HRV);Self-rated scores on the visual analog scale (VAS);Athlete Burnout Questionnaire (ABQ);Borg’s Rate of Perceived Exertion Scale (RPE).

HRV was measured using the Omegawave Athlete Real-time Functional State Comprehensive Diagnosis System (Omegawave Technologies LLC, Espoo, Finland, Ver. 3.6). Following the 30 min Stroop task, participants were assessed in a supine position with eyes closed and whole-body relaxation. Electrode pads were placed on the forehead and the left thenar eminence. Analysis of multidimensional HRV indicators determined whether participants reached the required level of mental fatigue [25,26,27,28].

HRV was measured using the Omegawave Athlete Real-time Functional State Comprehensive Diagnosis System (Omegawave Technologies LLC, Espoo, Finland, ver. 3.6). Following the 30 min Stroop task, participants were assessed in a supine position with eyes closed and whole-body relaxation. Electrode pads were placed on the forehead and the left thenar eminence.

VAS of mental fatigue was assessed using a 10 cm horizontal scale, anchored by “No fatigue at all” (0) and “Extremely fatigued” (10) [29]. Participants marked their perceived state on the line, and the distance (cm) from the start point provided a quantitative score. Participants completed three VAS ratings immediately before and after the Stroop task:VAS-MO: Motivation for upcoming test.VAS-MF: Current mental fatigue.VAS-ME: Current mental effort.

The ABQ is a validated measure of athlete burnout [30]. Participants completed the ABQ once before and once after the Stroop task. The questionnaire employs a Likert 5-point scale (1 = “Strongly Disagree” to 5 = “Strongly Agree”) across three dimensions: Emotional/Physical Exhaustion, Reduced Sense of Accomplishment, and Sport Devaluation. Scoring considers both the uniqueness and integration of these dimensions. This study employed the psychological fatigue calculation formula integrated by Zhang Liancheng et al. [31] for a comprehensive, objective analysis.

RPE was measured using the modified Borg’s RPE CR10 scale (0 = “No fatigue” to 10 = “Maximum extreme effort”). Participants rated their fatigue level immediately before and after the mental fatigue induction protocol [32].

### 2.3. Data Processing and Analysis

All experimental data are presented as mean (M) ± standard deviation (SD). Statistical analyses were performed using SPSS ver. 22.0 software (IBM, Chicago, IL, USA), with the significance level set at α = 0.05.

The normality of the data distribution was assessed using the Shapiro–Wilk test. Where the test result indicated normality (*p* ≥ 0.05), parametric tests (paired samples *t*-test) were used for analysis. Where the data deviated significantly from normality (*p* < 0.05), non-parametric tests (Wilcoxon signed-rank test) were employed. The post hoc effect size was calculated using Gpower Software, (ver. 3.1.9.7, Dusseldorf, Germany) yielding an ES = 0.35 (α = 0.05, critical t = 1.89).

## 3. Results

The specific time-domain and frequency-domain indicators of the subjects are shown in Table 1 and Table 2 below.

Mental fatigue significantly altered autonomic regulation, evidenced by HRV changes:

Parasympathetic suppression: HFnorm and RMSSD decreased (*p* < 0.05). Sympathetic dominance: LF/HF ratio increased five-fold (*p* < 0.05). Reduced overall variability: SDNN decreased (*p* < 0.05). LFnorm anomaly: Unexpectedly decreased (*p* < 0.05), suggesting altered physiological responses under fatigue.

### 3.1. Visual Analog Scale

Paired sample *t*-tests indicated that after completing the Stroop task, subjects’ motivation to exercise significantly decreased (*p* = 0.001), while mental fatigue level (*p* < 0.001) and mental effort level (*p* = 0.002) significantly increased (Table 3).

### 3.2. Rating of Perceived Exertion Scale Data

A significant difference in subjective mental fatigue before and after completing the Stroop task (t = 5.814, df = 7, *p* = 0.001) (Table 4).

### 3.3. Athlete Burnout Questionnaire

Paired sample *t*-tests indicated significant differences before and after completing the Stroop task for Emotional/Physical Exhaustion (*p* = 0.007), Reduced Sense of Accomplishment (*p* = 0.007), Sport Devaluation (*p* = 0.006), and overall burnout level (*p* = 0.002) (Table 5).

### 3.4. Loughborough Soccer Shooting Test Score

Table 6 shows the comparative analysis of the scores of the eight subjects before and after mental fatigue induction in this Loughborough Soccer Shooting Test. After normality testing and the corresponding pre- and post-data for each group, a paired *t*-test analysis was used for each data group based on the correlation coefficient. The analysis showed that after undergoing the 30 min Stroop task, the subjects’ left foot test scores (*p* = 0.013), right foot test scores (*p* = 0.001), and total scores (*p* < 0.001) significantly decreased.

### 3.5. Kicking Timing Before and After Mental Fatigue

Results of kicking timing are reported in Figure 1. FL/KF: Post-test time (0.90 s) was slightly longer than pre-test (0.83 s), an increase of approximately 8.8%. FL/BC: Post-test time (1.22 s) was slightly longer than pre-test (1.16 s), an increase of approximately 5.2%. FPRON: Post-test time (2.99 s) was basically the same as pre-test (2.96 s), only increasing by about 0.9%. TCT: Post-test time range (1.22–1.36 s) was overall shifted later compared to pre-test (1.16–1.29 s). Although the duration of all four events increased after mental fatigue, after paired *t*-tests, the *p*-values were all greater than 0.05, indicating no statistically significant difference. These results agree with a previous study, which showed that the average shot sequence time tended to be slower in the mental fatigue condition but failed to find a significant difference [15]. Using 2D sampling could have introduced significant sources of errors, albeit the movement is essentially mono-planar [25].

## 4. Discussion

### 4.1. Heart Rate Variability

HRV is a recognized indicator of mental fatigue [26,27]. Specifically, the LF/HF ratio reflects sympathetic nervous system activity and stress response. An increase in LF/HF indicates enhanced central nervous system regulation of the cardiovascular system, potentially linked to the stress response induced by elevated cognitive load [28,29]. HFnorm represents the regulatory influence of the parasympathetic nervous system on the heart rate. A decrease in HFnorm suggests suppression of the parasympathetic “rest-and-digest” function, signifying a heightened metabolic state [30]. Similarly, RMSSD primarily reflects the rapid parasympathetic (vagal) modulation of heart rate. Following mental fatigue induction, we observed a decrease in RMSSD, a finding consistent with other studies in a non-sport population [33]. RMSSD typically decreases under acute stress. Research indicates reductions of 15–25% following Stroop tasks, aligning with the decrease in HFnorm (indicating parasympathetic inhibition) activity, as reported in another non-sport population [34]. Studies that considered the same variables in soccer players are limited. A recent study in Chinese soccer players showed similar results [35], confirming our findings. During mental fatigue, concurrent sympathetic activation and parasympathetic withdrawal may partially counterbalance each other, potentially resulting in non-significant changes in SDNN; however, pronounced sympathetic dominance can lead to SDNN decreases. Our results, although the sample is limited, confirmed that mental fatigue induction significantly increased the LF/HF ratio and significantly decreased HFnorm.

### 4.2. Visual Analog Scale and Rating of Perceived Exertion

This study employed a combination of subjective and objective measures to evaluate and confirm successful mental fatigue induction in participants [36]. Given the specific need to verify induction after the 30 min Stroop task [37], the VAS was utilized, with VAS-MF (mental fatigue) serving as the primary subjective indicator. Data analysis revealed that post-task, participants exhibited significantly decreased VAS-MO (mental orientation) scores alongside significantly increased VAS-MF and VAS-ME (mental energy) scores. These VAS changes collectively indicate that the 30 min Stroop task successfully induced mental fatigue in all eight subjects.

Furthermore, participants completed the RPE scale in a resting state following the Stroop task, ensuring physiological stability. Analysis of pre- and post-task RPE scores showed a significant increase in self-perceived fatigue levels, despite the absence of high-load physical exercise before or after induction. This rise in RPE provides additional subjective evidence confirming the successful induction of mental fatigue. However, these changes induced by fatigue did not influence the timing of movement.

### 4.3. Athlete Burnout Questionnaire

This study revealed significant changes in participants’ psychological state following mental fatigue induction [35]. Scores for overall psychological fatigue, Emotional/Physical Exhaustion, Reduced Sense of Accomplishment, and Sport Devaluation all decreased significantly. Prior to induction, five participants (62.5%) exhibited mild psychological fatigue, while three (37.5%) exhibited severe fatigue. Post-induction, this shifted markedly: only one participant (12.5%) remained in the mild fatigue category, while seven (87.5%) reported severe psychological fatigue. Furthermore, the weighted total scores for these dimensions demonstrated a clear upward trend.

These findings, the significant decreases in specific burnout dimension scores, the pronounced shift towards severe fatigue classification, and the rising weighted scores—collectively indicate that the 30 min Stroop task successfully induced a state of mental fatigue. Consequently, participants experienced significant burnout across all three measured dimensions: Emotional/Physical Exhaustion, Reduced Sense of Accomplishment, and Sport Devaluation.

### 4.4. The Loughborough Soccer Shooting Test

The LSST, a validated measure of soccer shooting skill with established reliability and validity [14,22,23], was employed in this study. Following mental fatigue induction, shooting scores for both the left and right feet decreased significantly among the eight participants. This decline indicates that mental fatigue significantly impaired shooting accuracy in collegiate soccer players, a finding consistent with previous research [15,38].

Notably, the score fluctuation was greater for the non-dominant (left) foot. While non-dominant foot technique (encompassing power, accuracy, and coordination) is inherently less stable than the dominant foot [39], mental fatigue appears to further diminish control over the non-dominant limb, leading to a more pronounced drop in left-foot scores. This observation aligns with the proposed neural mechanisms of mental fatigue.

Furthermore, these results corroborate prior findings indicating that during moderate-to-high intensity exercise, attention is directed inward, requiring increased cognitive effort when movements demand greater cognitive control [40]. The neurophysiological basis for this phenomenon was deeply discussed before [41]. Under mental fatigue, limited attentional resources are likely prioritized for the dominant foot, reducing the neuromuscular control efficiency for the non-dominant foot. This diminished [41] control consequently leads to significant declines in non-dominant foot shooting accuracy and power stability [40]. The inhibitory effect of fatigue on motor unit recruitment in non-dominant limbs may also contribute to this exacerbated decline.

### 4.5. Events Duration (Kicking Timing)

The function of the support leg during shooting is not only to provide stable support but also to effectively maintain body balance, thereby creating more favorable conditions for the swinging leg to generate power [42]. Biomechanical analysis of the inside instep shot identifies the swing time of the kicking leg, the duration of ball contact, and the transfer of impact momentum as key factors influencing this technique.

In this study, high-speed cameras were used to analyze participants’ shooting actions frame by frame. The analysis focused on changes in the durations of four specific events before and after fatigue. Previous research has found that the duration of these events significantly increases following fatigue [15,43]. However, our experiment did not detect statistically significant changes in either the event durations. The higher cognitive load required to perform the LSST worsens LSST scores, while the timing, which has a lower cognitive load, has not been affected by mental fatigue. This lack of change in timing can also be explained by the limited sample size, lack of task learning, and measurement errors introduced by the 2D analysis, which can have impaired the results. For example, we recorded only one shot, and the variability among different shots could be high. Also, we neglected the contribution of the upper body and the inter-subject variability.

## 5. Conclusions

Although we are aware of the limitations of our study, which reside primarily in the sample size, we can draw some conclusions that can be useful for further experimental studies on the topic. This study confirms that a 30 min Stroop task can effectively induce mental fatigue in collegiate soccer players, evidenced by significant autonomic changes, subjective reports, and increased psychological burnout. Crucially, mental fatigue can impair shooting accuracy, particularly for the non-dominant foot, aligning with attentional depletion and reduced neuromuscular efficiency. Although biomechanical event durations and joint angles remained unchanged—contrary to prior findings—the overall results show that mental fatigue can have a detrimental impact on both psychophysiological state and soccer-specific performance. This finding must make the players and coaches aware of avoiding intensive mental-fatiguing tasks before a match (for example, intensive use of computers and phones) and underscore the need to address cognitive fatigue in training to maintain skill accuracy and mental resilience. Progressive habituation to mentally challenging tasks could help to sustain more accurate performance. Further direction of research can address the development and evaluation of specific programs of mental fatigue training for soccer players and the efficacy of educational interventions about the proper use of electronic devices for entertainment.

## Figures and Tables

**Figure 1 sports-13-00259-f001:**
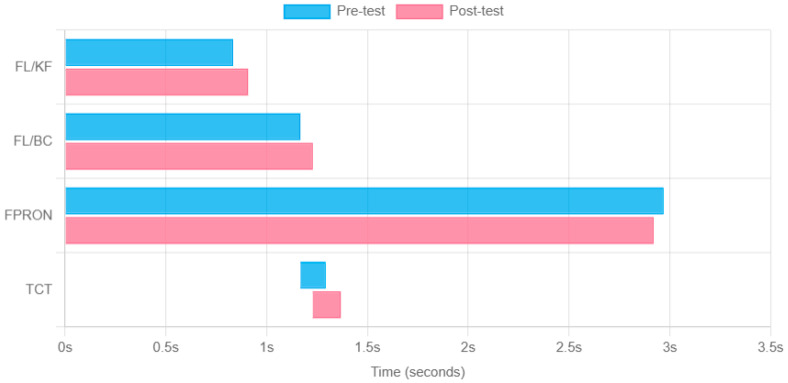
Event durations before and after mental fatigue. FL/KF: Time from landing of the supporting foot to maximum knee flexion. FL/BC: Time from landing of the supporting foot to ball contact. FPRON: Time from landing of support foot to maximum foot inversion. TCT (Total Contact Time): Total time from landing of support foot to ball contact.

**Table 1 sports-13-00259-t001:** HRV time-domain indicators.

Time-Domain Indicator	Baseline	Post-Mental Fatigue
RMSSD	55.41 ± 22.66	34.41 ± 21.12 *
SDSD	62.23 ± 16.32	40.28 ± 22.39 *
SDNN	61.58 ± 21.71	39.61 ± 22.13 *

RMSSD = Root Mean Square of Successive Differences between adjacent heartbeats; SDSD = Standard Deviation of Successive Differences between adjacent heartbeats; SDNN = Standard Deviation of Normal-to-Normal (NN) intervals; * indicates comparison with baseline, *p* < 0.05. ES = 0.35.

**Table 2 sports-13-00259-t002:** HRV frequency-domain indicators.

Frequency-Domain Indicator	Baseline	Post-Mental Fatigue
LFnorm	453.26 ± 69.39	137.78 ± 89.64 *
HFnorm	905.28 ± 297.43	375.68 ± 232.56 *
LF/HF	0.99 ± 0.42	5.94 ± 0.54 *

LFnorm = Normalized Low-Frequency power; HFnorm = Normalized High-Frequency power; LF/HF = Low-Frequency to High-Frequency ratio; * indicates significant differences from baseline, *p* < 0.05. ES = 0.35.

**Table 3 sports-13-00259-t003:** Changes in VAS pre- and post-mental fatigue.

Scale	Time Point	Score	Paired Sample *t*-Test (*p*)
VAS-MO	Pre-task	7.50 ± 1.69	0.001
	Post-task	3.38 ± 1.69	
VAS-MF	Pre-task	2.50 ± 1.07	<0.001
	Post-task	7.38 ± 1.06	
VAS-ME	Pre-task	2.75 ± 1.58	0.002
	Post-task	6.75 ± 2.05	

VAS = visual analog scale (points). ES = 0.35.

**Table 4 sports-13-00259-t004:** RPE rating pre- to post-mental fatigue.

Scale	Time Point	Score	Paired Sample *t*-Test (*p*)
RPE	Pre-task	3.00 ± 9.26	0.001
	Post-task	6.25 ± 1.389	

RPE = Rating of Perceived Exertion Scale. ES = 0.35.

**Table 5 sports-13-00259-t005:** Changes in Athlete Burnout Questionnaire scores.

Dimension	Time Point	Score	Paired Sample *t*-Test (*p*)
Emotional/Physical Exhaustion	Pre-task	13.00 ± 3.02	0.007
	Post-task	16.13 ± 2.03	
Reduced Sense of Accomplishment	Pre-task	15.25 ± 1.39	0.007
	Post-task	17.88 ± 1.80	
Sport Devaluation	Pre-task	10.50 ± 3.46	0.006
	Post-task	16.25 ± 3.81	
Burnout Level	Pre-task	38.75 ± 6.69	0.002
	Post-task	50.25 ± 6.41	

ES = 0.35, α = 0.05, critical t = 1.89.

**Table 6 sports-13-00259-t006:** LSST scores.

Foot	Time Point	Score	Paired Sample *t*-Test (*p*)
Left	Pre-task	11.38 ± 3.85	0.013
	Post-task	7.25 ± 1.67	
Right	Pre-task	12.00 ± 2.00	0.001
	Post-task	9.00 ± 2.33	
Total	Pre-task	11.69 ± 2.98	<0.001
	Post-task	8.56 ± 2.76	

## Data Availability

Data are available at Figshare repository: https://doi.org/10.6084/m9.figshare.29369597.v1.

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
