# Peer review of "The Impact of Mental Fatigue on the Accuracy of Penalty Kicks in College Soccer Players"

_sports, 2025, doi:10.3390/sports13080259_

Round 1
Reviewer 1 Report
Comments and Suggestions for Authors
This study aimed to investigate the effect of mental fatigue on shooting accuracy in collegiate male soccer players. While the scientific approach to analyzing the impact of mental fatigue on performance is interesting, the extremely limited sample size warrants caution in interpreting the results. Detailed review comments are as follows:
- Abstract: The description of the results should include relevant statistical values (e.g., p-values, means ± SD) for clarity and precision.
- Methods: It is difficult to generalize the results from only 8 participants. At the very least, effect sizes should be reported to support the validity of the study's claims. A summary table outlining the physical characteristics of the participants should be added.
- Results: Effect sizes should be presented for each table to enhance the interpretability of the findings.
- Discussion:
1) The conclusions drawn in this study tend to be overstated. The wording should be moderated. Furthermore, due to the basic level of statistical analysis used, the interpretation of relationships and causality among the results is insufficient.
2) Despite the extremely limited sample size, many expressions in the discussion are overly definitive.
3) The discussion should better integrate findings from previous studies to provide context and comparison.
4) The practical implications of the results—specifically how they can be applied in real-world sports settings—should be discussed.
5) The limitations mentioned above should be explicitly addressed in the discussion, along with suggestions for future research directions.
Author Response
Dear Reviewer nr.1
Thank you very much for reviewing our paper. We appreciate very much.
In red color a point-by-point answer.
Abstract: The description of the results should include relevant statistical values (e.g., p-values, means ± SD) for clarity and precision.
We incorporate the highlight of the results in the abstract. However, is not possible to report in extenso all the results, or the abstract will be extremely long.
Methods: It is difficult to generalize the results from only 8 participants. At the very least, effect sizes should be reported to support the validity of the study's claims. A summary table outlining the physical characteristics of the participants should be added.
Results: Effect sizes should be presented for each table to enhance the interpretability of the findings.
We added the mean ES to each table and added in the methods : Post-hoc effect size were calculated with Gpower Software and gave and ES = 0.35 (α = 0.05, critical t = 1.89).
Discussion:
1) The conclusions drawn in this study tend to be overstated. The wording should be moderated. Furthermore, due to the basic level of statistical analysis used, the interpretation of relationships and causality among the results is insufficient.
We modified to a more probabilistic language
2) Despite the extremely limited sample size, many expressions in the discussion are overly definitive.
We modified to a more probabilistic language
3) The discussion should better integrate findings from previous studies to provide context and comparison.
We modified accordingly
4) The practical implications of the results—specifically how they can be applied in real-world sports settings—should be discussed.
Added, This finding must make aware the players and coaches to avoid intensive mental-fatiguing tasks before a match and underscore the need to address cognitive fatigue in training to maintain skill accuracy and mental resilience. Progressive habituation to mental challenging tasks could help to sustain more accurate performance. A further direction of research can address the development and evaluation of specific programs of mental fatigue training for soccer players. We also added a recent reference to a similar study in soccer [35].
5) The limitations mentioned above should be explicitly addressed in the discussion, along with suggestions for future research directions.
Added. “Albeit we are aware of the limitation of our study, which resides primarly in the sample size, we can draw some conclusions, which can be useful for further experimental studies on the topic.

Reviewer 2 Report
Comments and Suggestions for Authors
Introduction
This section is well structured and provides a comprehensive overview of the concept of mental fatigue and its effects on sports performance. The references are appropriate and relevant. The authors point out the shortcomings of previous studies in the field of Asian population. However, the text is repetitive in places, particularly in the references to the Stroop test, and there is a lack of clear introduction of the research hypotheses. State the main hypotheses of the study clearly at the end of the introduction and reduce the repetitive phrases to make the text more focused.
Methods
This section describes the study design,instruments, and the protocol in a detailed and transparent manner. The multidimensional measurement approach is a strength. The study sample is extremely small, which significantly limits the generalizability of the results. The measurement and processing procedures for biomechanical data (timing indicators) are less detailed, and reliability aspects of motion analysis are particularly lacking. Provide a more detailed methodological description of the processing of timing data and to emphasize the research limitations arising from the small sample size.
Results
The presentation of the timing indicators is rather descriptive and their statistical interpretation is less detailed Their practical significance should be explained. The tables and figures may be difficult to understand while they do not always clearly explain the meaning of the indicators. A more in-depth statistical analysis of the biomechanical results and a more detailed presentation of the practical interpretation of the results would be beneficial.
Discussion
The discussion section reflects well on the results. However, the limitations of the small sample size are underrepresented. Also, some conclusions are overly generalised. The explanation for the consistency of the kicking timing results does not address other possible causes (e.g., task learning, measurement errors). Explain the limitations more and put an emphasis on the need for further studies with a larger sample.
Conclusions
The conclusions summarize the most important results, namely the successful induction of mental fatigue and its negative effect on shooting accuracy. Practical aspects are also mentioned. However, the limiting factors (small sample size, laboratory setting) are not emphasized enough, and future research directions are only mentioned briefly. More specific practical recommendations and more detailed research directions would be great to be added.
Overall, the manuscript presents a well-designed and adequately conducted study focusing on a multidisciplinary investigation of the effects of mental fatigue on sports performance. Among the strengths, the complex measurement approach and the relevant research question must be emphasised. The main weakness is the small sample size and the resulting statistical and generalizability limitations. To improve the quality of the manuscript, I recommend:
- clear formulation of hypotheses,
- more detailed analysis of biomechanical results,
- more emphasis on research limitations,
- more precise formulation of practical and theoretical conclusions.
Author Response
Dear Reviewer nr. 2.
Thank you very much for your precious time in reviewing this paper.
We addressed all the point below.
Introduction
This section is well structured and provides a comprehensive overview of the concept of mental fatigue and its effects on sports performance. The references are appropriate and relevant. The authors point out the shortcomings of previous studies in the field of Asian population. However, the text is repetitive in places, particularly in the references to the Stroop test, and there is a lack of clear introduction of the research hypotheses. State the main hypotheses of the study clearly at the end of the introduction and reduce the repetitive phrases to make the text more focused.
We modified accordingly.
Methods
This section describes the study design,instruments, and the protocol in a detailed and transparent manner. The multidimensional measurement approach is a strength. The study sample is extremely small, which significantly limits the generalizability of the results. The measurement and processing procedures for biomechanical data (timing indicators) are less detailed, and reliability aspects of motion analysis are particularly lacking. Provide a more detailed methodological description of the processing of timing data and to emphasize the research limitations arising from the small sample size.
We added: “Timing is a basic measurement of the performance on the instep kicking in soccer, and was proposed as a simple, no-time consuming and easily understandable tool for biomechanical performance assessment using simple measures [24]. Thus, this method has the advantage of not requiring complex calculations and to be quickly usable on the soccer field. Pre-post mental fatigue times differences were computed
Results
The presentation of the timing indicators is rather descriptive and their statistical interpretation is less detailed Their practical significance should be explained. The tables and figures may be difficult to understand while they do not always clearly explain the meaning of the indicators. A more in-depth statistical analysis of the biomechanical results and a more detailed presentation of the practical interpretation of the results would be beneficial.
We added: “Timing is a basic measurement of the performance on the instep kicking in soccer, and was proposed as a simple, no-time consuming and easily understandable tool for biomechanical performance assessment using simple measures [24]. We also added:
These results are in agreement with a previous study which showed the average shot sequence time tended to be slower in the mental fatigue condition, but failed to find a significant differences [15]. Using 2D sampling, could have introduced significant sources of errors, albeit the movement is essentially monoplanar [25].
Discussion
The discussion section reflects well on the results. However, the limitations of the small sample size are underrepresented. Also, some conclusions are overly generalised. The explanation for the consistency of the kicking timing results does not address other possible causes (e.g., task learning, measurement errors). Explain the limitations more and put an emphasis on the need for further studies with a larger sample.
Added: However, our experiment did not detect statistically significant changes in either the event durations. This can be explained with the limited sample size, lack of task learning, and measurement errors introduced by the 2D analysis, which can have impaired the results.
Conclusions
The conclusions summarize the most important results, namely the successful induction of mental fatigue and its negative effect on shooting accuracy. Practical aspects are also mentioned. However, the limiting factors (small sample size, laboratory setting) are not emphasized enough, and future research directions are only mentioned briefly. More specific practical recommendations and more detailed research directions would be great to be added.
We added the limitation of small sample size and methodological sources of errors: Albeit we are aware of the limitation of our study, which resides primarily in the sample size, we can draw some conclusions, which can be useful for further experimental studies on the topic.
This finding must make aware the players and coaches to avoid intensive mental-fatiguing tasks before a match and underscore the need to address cognitive fatigue in training to maintain skill accuracy and mental resilience. Progressive habituation to mental challenging tasks could help to sustain more accurate performance. A further direction of research can address the development and evaluation of specific programs of mental fatigue training for soccer players.
Overall, the manuscript presents a well-designed and adequately conducted study focusing on a multidisciplinary investigation of the effects of mental fatigue on sports performance. Among the strengths, the complex measurement approach and the relevant research question must be emphasised. The main weakness is the small sample size and the resulting statistical and generalizability limitations. To improve the quality of the manuscript, I recommend:
c

Reviewer 3 Report
Comments and Suggestions for Authors
Dear authors,
A good research design that looked at the effects of the Stroop test on soccer performance. As the performance variables were not affected as much as the mental this study did show the importance of mental fatigue on soccer players perception and physical state that then in turned played a role on shooting accuracy. This study brings up a big question that was not addressed in the study that needs to be added to the discussion is how does one apply this knowledge. As a soccer player will not performs a Stroop test before a game, but they might sit at a computer or on their screens all day and this it will cause a mental fatigue that can have an impact on their performance. How does one knowing how mental fatigue affect performance go forward?
Author Response
Dear Reviewer nr. 3
Thank you very much for your time spent in reviewing our paper.
We modified significantly the paper in different parts of the writing and added some explanations and deepening. Specifically:
This finding must make aware the players and coaches to avoid intensive mental-fatiguing tasks before a match and underscore the need to address cognitive fatigue in training to maintain skill accuracy and mental resilience. Progressive habituation to mental challenging tasks could help to sustain more accurate performance. A further direction of research can address the development and evaluation of specific programs of mental fatigue training for soccer players and the efficacy of educational interventions about a proper use of electronic devices for entertainment
A good research design that looked at the effects of the Stroop test on soccer performance. As the performance variables were not affected as much as the mental this study did show the importance of mental fatigue on soccer players perception and physical state that then in turned played a role on shooting accuracy. This study brings up a big question that was not addressed in the study that needs to be added to the discussion is how does one apply this knowledge. As a soccer player will not performs a Stroop test before a game, but they might sit at a computer or on their screens all day and this it will cause a mental fatigue that can have an impact on their performance. How does one knowing how mental fatigue affect performance go forward?

Round 2
Reviewer 1 Report
Comments and Suggestions for Authors
I believe that the manuscript has been appropriately revised in accordance with my comments, and the authors have adequately addressed the concerns I raised. This paper is suitable for publication in this journal. However, I recommend a final check for minor errors and language refinement to ensure the overall quality of the journal and the manuscript.
Furthermore, as the authors have pointed out, the abstract has become lengthy; therefore, I recommend again summarizing only the key findings more concisely in the abstract. I also agree that it is not necessary to include all p-values in the abstract. However, it is important to standardize the formatting of p-values. For example, expressions such as p = 0.002 and p = 0.013 should be consistently presented as p < 0.001, p < 0.05, and so on.
Comments on the Quality of English LanguageI believe that the manuscript has been appropriately revised in accordance with my comments, and the authors have adequately addressed the concerns I raised. This paper is suitable for publication in this journal. However, I recommend a final check for minor errors and language refinement to ensure the overall quality of the journal and the manuscript.
Furthermore, as the authors have pointed out, the abstract has become lengthy; therefore, I recommend again summarizing only the key findings more concisely in the abstract. I also agree that it is not necessary to include all p-values in the abstract. However, it is important to standardize the formatting of p-values. For example, expressions such as p = 0.002 and p = 0.013 should be consistently presented as p < 0.001, p < 0.05, and so on.
Author Response
Dear Reviewer nr. 2, thank you very much for your suggestions and time spent to review this paper.
We have shortened the abstract and modified the reporting of p values.
We also checked the English language for errors.
Best Regards